# Serum-Derived Extracellular Vesicles from African Swine Fever Virus-Infected Pigs Selectively Recruit Viral and Porcine Proteins

**DOI:** 10.3390/v11100882

**Published:** 2019-09-20

**Authors:** Sergio Montaner-Tarbes, Myriam Pujol, Tamara Jabbar, Philippa Hawes, Dave Chapman, Hernando del Portillo, Lorenzo Fraile, Pedro J. Sánchez-Cordón, Linda Dixon, Maria Montoya

**Affiliations:** 1Innovex Therapeutics S.L., 08916 Badalona, Barcelona, Spain; smontaner@igtp.cat (S.M.-T.); hernandoa.delportillo@isglobal.org (H.d.P.); lorenzo.fraile@ca.udl.cat (L.F.); 2Departamento de Ciència Animal, Escola Tècnica Superior d’Enginyeria Agrària, Avenida Alcalde Rovira Roure, 191, 25198 Lleida, Spain; 3Faculty of Medicine, Universidad de Chile, Santiago 7591538, Chile; myriampujol@gmail.com; 4The Pirbright Institute, Ash Road, Pirbright, Woking, Surrey GU24 0NF, UK; tamara.jabbar@pirbright.ac.uk (T.J.); pippa.hawes@pirbright.ac.uk (P.H.); chapmand@MedImmune.com (D.C.); Pedro.Sanchez-Cordon@apha.gov.uk (P.J.S.-C.); linda.dixon@pirbright.ac.uk (L.D.); 5Centro de Investigaciones Biológicas (CIB-CSIC), Universidad Complutense de Madrid, Ramiro de Maeztu 9, 28040 Madrid, Spain

**Keywords:** extracellular vesicles, African swine fever virus, proteomic analysis

## Abstract

African swine fever is a devastating hemorrhagic infectious disease, which affects domestic and wild swines (*Sus scrofa*) of all breeds and ages, with a high lethality of up to 90–100% in naïve animals. The causative agent, African swine fever virus (ASFV), is a large and complex double-stranded DNA arbovirus which is currently spreading worldwide, with serious socioeconomic consequences. There is no treatment or effective vaccine commercially available, and most of the current research is focused on attenuated viral models, with limited success so far. Thus, new strategies are under investigation. Extracellular vesicles (EVs) have proven to be a promising new vaccination platform for veterinary diseases in situations in which conventional approaches have not been completely successful. Here, serum extracellular vesicles from infected pigs using two different ASFV viruses (OURT 88/3 and Benin ΔMGF), corresponding to a naturally attenuated virus and a deletion mutant, respectively, were characterized in order to determine possible differences in the content of swine and viral proteins in EV-enriched fractions. Firstly, EVs were characterized by their CD5, CD63, CD81 and CD163 surface expression. Secondly, ASFV proteins were detected on the surface of EVs from ASFV-infected pig serum. Finally, proteomic analysis revealed few specific proteins from ASFV in the EVs, but 942 swine proteins were detected in all EV preparations (negative controls, and OURT 88/3 and Benin ΔMGF-infected preparations). However, in samples from OURT 88/3-infected animals, only a small number of proteins were differentially identified compared to control uninfected animals. Fifty-six swine proteins (Group Benin) and seven proteins (Group OURT 88/3) were differentially detected on EVs when compared to the EV control group. Most of these were related to coagulation cascades. The results presented here could contribute to a better understanding of ASFV pathogenesis and immune/protective responses in the host.

## 1. Introduction

Currently, African swine fever (ASF) constitutes the biggest threat faced by the world pork industry. The causative agent, African swine fever (ASFV), is a large and complex double-stranded DNA virus belonging to the *Asfarviridae* family [1]. ASFV affects domestic pigs and wild boars as well as other African suids. Clinical signs during infection vary considerably from acute forms, with mortality rates up to 90–100% in domestic pigs and wild boar to inapparent infections in bushpigs (*Potamochoerus sp.*) and warthogs (*Phacochoerus sp*.). The latter are involved in the sylvatic cycle with a soft tick vector *Ornithodoros sp.* [2]. The mature viral particle has an icosahedral morphology with a size range of 180–200 nm in diameter and is composed of multiple layers (nucleoprotein core, core shell, inner envelope, viral capsid and external envelope when the virus egresses from the cell) [3]. The genome is a double-stranded DNA molecule located in the nucleoprotein core, with approximately 170 to 193 kbp codifying for 151 to 167 open reading frames (ORFs). Most differences in genome size are associated with the gain or loss of members of multigene families (MGF) [4,5]. Mature viral particles infect primarily monocytic/macrophages by clathrin-mediated endocytosis or macropinocytosis, and egress is completed by the transport of mature viral particles by the microtubule network and finally budding from the plasma membrane [6,7]

In 2007, ASFV was reintroduced to Europe after eradication in the mid-1990s from Spain and Portugal, although the disease remained in Sardinia [8]. ASF-positive wild boar in Belgium and Eastern Europe represent a significant reservoir that may hinder eradication [9,10]. The seriousness of the ASF threat is exemplified by the first cases of ASFV in August 2018 in the world’s largest pig producer, China [11]; the disease has now spread to all mainland Chinese provinces and reached Vietnam, Cambodia, Laos and North Korea in 2019. Thus, the development of control and vaccination strategies to prevent ASF has become even more important [12,13,14].

Developing a safe and effective DIVA (differentiate infected from vaccinated animals) vaccine against ASFV is currently one of the main issues in animal health. Several attempts have been made to develop a vaccine for ASFV using inactivated vaccines, subunit approaches and live attenuated viruses, including gene deleted viruses [15] ranging from partial to some protection with some safety concerns [16,17,18]. One example is the non-virulent, non-hemadsorbing Portuguese isolate OURT 88/3 which belongs to the genotype I [19]. When combined with a boost of a closely related hemadsorbing Portuguese isolate (OURT 88/1), it can induce homologous and heterologous protection from 60 to 100% with almost no clinical signs or detectable viremia [17]. Another attenuated strain is the deletion mutant Benin ΔMGF, which was derived from the wild-type virulent genotype I isolate Benin97/1 by the deletion or interruption of several genes that inhibit the type I IFN response (MGF360-10L, 11L, 12L, 13L, 14L and MGF530/505-1R, 2R and 3R and MGF360-9L and MGF530/505-4R). This deletion mutant is capable of inducing homologous protection against the parental isolate; however, clinical signs are observed, and viremia is detected in infected pigs [20]. Nevertheless, several biosafety problems could arise when immunization with attenuated viral strains is translated to the field. Therefore, research into new DIVA “virus-free strategies” for vaccination is a priority in the field [16].

On the other hand, extracellular vesicles (EVs) have proven to be a promising new vaccination platform for veterinary diseases in situations in which conventional approaches have not been completely successful, such as PRRSV [21,22,23,24]. EVs are small, round vesicles of 50–400 nm in diameter which are secreted by different cell types, and they are classified as microvesicles if formed by the direct budding of the plasma membrane or exosomes if derived from late endosome trafficking and multivesicular body formation before final release to the extracellular space [25,26]. Importantly, exosome-like vesicles share the same pathway of formation as some viruses within multivesicular bodies and viral factories, whereby viruses send proteins and nucleic acids to the extracellular space, which later on can trigger immune responses against or in favor of viral replication [27,28,29].

EVs have been implicated in different biological processes, such as spreading viral infection or modulating immune responses in the host by means of exploiting EV formation pathways to include viral proteins, causing the activation of cellular and antibody-mediated immune pathways [27,29,30]. One example is the VP40 of Ebola virus, which can induce apoptosis in the recipient cells and is more effective in monocytes treated with EVs with a higher content of VP40 [31,32]. In contrast, during dengue virus infection, there is a high production of EVs containing IFN-inducible transmembrane proteins 1, 2 and 3, which activates an antiviral state in different cells and contributes to the IFN-induced inhibition of viral replication [33]. EVs produced during other viral infections such HIV-1 have ambiguous roles depending of the type of EVs and the cell type producing those vesicles; for example, EVs released by CD4+ T cells mediate the CD4-dependent inhibition of HIV-1 infection in vitro. On the other hand, a large proportion of EVs released by HIV-1-infected cells contain gp120 protein, which is supposed to play an important role by increasing viral infectivity in human lymphoid tissue [28]. Finally, EVs could play a protective role in avoiding the interaction between viral particles and extracellular molecules, such as by neutralizing antibodies. New evidence has also suggested that viruses (noroviruses and rotaviruses) form clusters that are included into vesicles, generating a more infectious unit containing more than one infectious particle that can resist the extracellular environment [34].

Given the role of extracellular vesicles in inducing immunological responses in different organisms and the context of diseases, together with the fact that there is a search for new vaccine approaches for ASFV, the main objective of this work was to characterize extracellular vesicles from serum from ASFV-infected animals. Following this line of thought, two different ASFV viruses (OURT 88/3 and Benin ΔMGF) were chosen when pigs survived infection for a sampling period. These correspond to a naturally attenuated virus and a deletion mutant, respectively. This experimental procedure allowed us to determine possible differences in swine and viral protein content on EV-enriched fractions that could contribute to the better understanding of ASFV pathogenesis and immune/protective responses in the host.

## 2. Materials and Methods

### 2.1. Cells and Viruses

Both the low-virulence non-hemabsorbing genotype I ASFV isolate OURT88/3 and attenuated deletion mutant Benin ΔMGF, obtained from virulent genotype I isolate Benin97/1, were grown in primary macrophage cultures derived from bone marrow. The preparation of viruses and virus titration was carried out as previously reported [19,20,35]. Results are presented as the amount of virus infecting 50% of the macrophage cultures (TCID50/mL).

### 2.2. Experimental Design, Sampling, Clinical Evaluation and Viremia

Experiments were conducted in SAPO4 high containment facilities at The Pirbright Institute and regulated by the Animals (Scientific Procedures) Act UK 1986 (Project License 70/8852, 05/04/2016). Samples were taken from a larger experiment from day 0 to day 24 following the 3R concept for minimizing animal use. Large White and Landrace crossbred female pigs, of 9 to 10 weeks old (21–25 kg), from a high health status herd were used. Pigs were separated into two groups of six pigs each. In group A (pigs A1 to A6), animals were immunized by the intramuscular route (IM) in the neck muscles with 1 mL containing 10^4^ TCID50/mL of OURT88/3 isolate, whereas in group B (pigs B1 to B6), animals were immunized using the same route and dose with the deletion mutant Benin ΔMGF. The immunization day was defined as day 0 (0 dpi). EDTA blood and serum samples from the jugular vein were collected from all pigs at day 7 and 24 pi. Rectal temperatures and clinical signs were monitored daily as described [17]. DNA was extracted from blood samples taken at days 7 and 24 post infection and analyzed for ASFV genome detection by quantitative PCR (qPCR) [36].

### 2.3. Extracellular Vesicle Isolation: Size-Exclusion Chromatography

The isolation of serum-derived EVs by size exclusion chromatography (SEC) was performed as previously described with swine sera samples from 7 dpi and 24 dpi [21]. Briefly, Sepharose CL-2B (Sigma-Aldrich, St. Louis, MO, USA) was packed in 10 mL syringes to a final volume of 10 mL and equilibrated with PBS-citrate 0.32% (*w/v*). Frozen serum samples were thawed, centrifuged at 500× *g* for 10 min at room temperature to remove cellular debris, and 2 mL aliquots were loaded to each column. The collection of 20 fractions of 0.5 mL each started immediately, using PBS-citrate as the elution buffer. The protein content of each fraction was analyzed using a Bradford protein quantification assay according to the manufacturer’s instructions (Pierce BCA protein quantification assay kit, Thermo-Fisher, Waltham, MA, USA).

### 2.4. Flow Cytometry Analysis of Molecular Markers Associated with Extracellular Vesicles

A bead-based assay for the detection of classical exosome markers, CD63 and CD81, and two new markers (CD5 and CD163) was used with some modifications to allow the use of this protocol in a BSL-3 environment. Briefly, EV-enriched fractions were coupled with Aldehyde/Sulfate Latex Beads, 4% *w/v*, at 4 µm (Invitrogen, Carlsbad, CA, USA) and then blocked with PBS 1×/BSA 0.1% (Sigma-Aldrich)/NaN3 0.01% (Sigma-Aldrich). Fractions were incubated in microtest conical-bottom 96-well plates for 30 min at 4 °C with anti-CD63 (Clone H5C6) and anti-CD81 (Clone JS-81) antibodies (BD Biosciences, San Jose, CA, USA) at 1:100 dilution, anti-CD5 (clone PG114A, Kingfisher Biotech, St Paul, MN, USA) at 1:200 or anti-CD163 (clone 2A10 gently given by Dr. Javier Dominguez). After washing, a 1:100 dilution of secondary antibody anti-mouse FITC (Southern Biotech, Birmingham, AL, USA) was incubated for 30 min at 4 °C. After the removal of unbound secondary antibodies by centrifugation and washing with PBS/BSA, beads were suspended with 4% paraformaldehyde for 30 min to inactivate any possible contamination with ASFV particles. Then, the plate was centrifuged and beads resuspended in PBS and analyzed by flow cytometry using MACSQuant Analyzer 10 equipment (Miltenyi Biotec, Bergisch Gladbach, Germany). The median fluorescence intensity (MFI) and bead count data were obtained by FlowJo vX for PC TreeStar analysis software for every sample-reading file.

### 2.5. Detection of Viral Proteins on the Surface of Extracellular Vesicles Using Bead-Based Assay

A bead-based assay for the detection of viral proteins was used to phenotypically identify size exclusion chromatography fractions containing EVs and viral proteins [21]. Briefly, EV-enriched fractions were coupled with aldehyde/sulfate latex beads, 4% *w/v*, 4 µm (Invitrogen) and then blocked with PBS 1×/BSA 0.1% (Sigma-Aldrich)/NaN3 0.01% (Sigma-Aldrich). Fractions were incubated in microtest round-bottom 96-well plates for 30 min at 4 °C with anti-p30, anti-p54 and anti-p72 antibodies (The Pirbright Institute) [17,37,38] at 1:100 dilution. After washing, a 1:100 dilution of secondary antibody anti-mouse-FITC (Southern Biotech) was incubated for 30 min at 4 °C. After the removal of unbound secondary antibodies by centrifugation, beads were resuspended in 4% paraformaldehyde for 30 min to inactivate any possible contamination with ASFV particles. Then, samples were centrifuged and beads resuspended in PBS and analyzed by flow cytometry using MACSQuant Analyzer 10 equipment (Miltenyi Biotec). The median fluorescence intensity (MFI) and bead count data were obtained by FlowJo vX for PC TreeStar analysis software for every sample-reading file.

### 2.6. Transmission Electron Microscopy and Negative Staining

Ten microliters of each sample were placed on Formvar-coated, glow discharged copper grids for one minute. After removing excess sample with filter paper, the grids were briefly placed on droplets of distilled water and the excess removed. The grids were placed on droplets of 3% aqueous uranyl acetate for one minute before excess staining was removed and the grids were allowed to dry. Samples were imaged using a FEI T12 TEM at 100 kV with a Tietz F214 camera. For size distribution, all images were evaluated using ImageJ software and plotted in size-range percentages.

### 2.7. Mass Spectrometry

Extracellular vesicle fractions from size exclusion chromatography were heat-inactivated at 65 °C for 2 h to inactivate any ASFV and sent to the Biological Mass Spectrometry Facility of the University of Sheffield. Liquid chromatography (nanoLCULTRA-EKSIGENT) followed by mass spectrometry (nanoLC-MS/MS) was performed on an LTQ Orbitrap Velos (Thermo Fisher). Briefly, samples were reduced with 10 mM DTT (Dithiothreitol), alkylated with 55 mM iodoacetamide and precipitated by 10% TCA (trichloroacetic acid). After washing with acetone, 2 μL of 8 M urea was added and samples were brought to a final concentration of 1.6 M urea. One microgram of trypsin (*Sus scrofa*) was added and digestions were carried out overnight at 37 °C. The reaction was stopped with 1% formic acid. The amount of sample submitted to mass spectrometry analyses was based on nanoparticle tracking analysis (see below) and ranged from 9.8 × 10^7^ to 3.9 × 10^8^ particles/mL among all samples analyzed. MS/MS was performed in the LTQ using the data-dependent dynamic exclusion of the top 20 most intense peptides using a repeat count of 1, a repeat duration of 30 s, an exclusion list size of 500 and exclusion list duration of 30 s as parameters. The top 20 most intense peptides were isolated and fragmented by low-energy CID at 35% collision energy.

### 2.8. Database Search and Protein Identification

Raw spectral data were investigated against a custom database compiled in FASTA format to upload it into Maxquant 1.6. The databases were obtained from UniProtKB and contained complete and partial sequences of ASFV (4125 sequences, 806 reviewed) and *Sus scrofa* (40,713 complete proteome sequences). The sequences for the default contaminant database was also included in the search carried out with Maxquant 1.6 software. Contaminants and proteins identified only by site modification were filtered out from the list. Proteins found in all groups were scored positive if they had at least two unique peptides and a 1% false discovery rate (FDR) for protein and peptide identification. In ASFV protein analyses, the identification also included the criteria of being present on infected samples and absent in control samples to avoid false identification hits due to the small number of proteins in the ASFV UniprotKB database. ASFV proteins were evaluated only in terms of their presence/absence while swine proteins were compared using a relative quantification approach.

### 2.9. Statistical Analysis

Pig protein hits were evaluated using Perseus v1.6. Contaminants, reversed identified hits and those identified by site modification proteins were filtered out of the analysis. The matrix was reduced by eliminating proteins identified with less than 1 unique peptide and 1% FDR. Then, sample groups of controls and infected swine proteins identified in EV-enriched preparations were compared using a two-sample Student’s *t*-test of the mean normalized intensity (LFQ) by the permutation-based FDR method. Differentially expressed proteins were marked as significant if *p* < 0.05 (red) and *p* < 0.01 (volcano plot above threshold). After filtering, proteins of each group were compared in a Venn diagram using Venny 2.1 [39] software to determine which proteins were unique and shared among groups.

### 2.10. Gene Ontology Analysis

Proteins found to be statistically significant by the Student’s *t*-test were evaluated for gene ontology. For this, identified hits were examined using the Database for Annotation, Visualization and Integrated Discovery (DAVID) v6.8 [40] to identify significant enrichment in terms of biological processes, molecular functions and cellular components using the -log (*p* value) obtained previously, and results were expressed in categories belonging to each enrichment term. Information about protein interactions between significantly enriched proteins was retrieved by STRING software [41].

## 3. Results

### 3.1. Viremia and Clinical Signs of ASFV

As reported previously [42], OURT88/3-immunized pigs did not show changes in rectal temperatures (Figure 1). However, except for pig B6, a transient increase in rectal temperature above 40.5 °C for 1 or 3 days was observed in all pigs immunized with Benin ΔMGF between days 3 and 5 post immunization. Nevertheless, no other clinical signs were observed along with increased temperatures.

Blood samples were taken at day 7 and 24 for EV purification. Those samples were tested for ASFV DNA content by quantitative PCR (qPCR) [36]. Due to possible virus replication at day 7 post immunization, those samples could not be used to evaluate the viral protein content of EVs, as viruses will mask the proteins associated with them. However, samples were checked at 14 and 24 dpi, and none of the pigs immunized with OURT88/3 (group A) showed detectable levels of virus genome in blood, while in all pigs immunized with Benin ΔMGF (group B), moderate levels of ASFV DNA were detected ranging from 2.85 × 10^4^ to 1.45 × 10^5^ copies at 14 dpi and 4.03 × 10^3^ to 2 × 10^4^ copies at 24 dpi [20]. As some EVs and viral particles have similar sizes, it was relevant to establish whether ASFV was present in extracellular vesicle-enriched fractions to avoid false identifications of viral proteins in further analyses. However, they could be used for the standardization and evaluation of swine protein cargo on EV-enriched fractions.

### 3.2. Extracellular Vesicles in ASFV-Infected Sera

Both naïve (Figure 2a–c) and ASFV-infected (Figure 2d–f) sera at 7 dpi were used to obtain EV-enriched fractions to standardize all the procedures. As described previously [21], swine EVs eluted in fractions 7–10 from the exclusion chromatography column displayed CD163 and CD81 high MFI values and exhibited a mean size distribution of 100–200 nm measured by TEM (Figure 2c,f). Both naïve (Figure 2a–c) and ASFV-infected (Figure 2d–f) sera at 7 dpi were used to obtain EV-enriched fractions to standardize all the procedures.

In addition, there was no difference in the elution profile in terms of protein and FACS for the molecular markers, independently of the experimental group (naïve or infected swine serum from both groups) and the virus used. However, TEM showed unidentified structures (Figure 1f, red arrows) in some samples of EV-enriched fractions from ASFV-infected animals independently of the virus strain used at 7 dpi (Figure 2f) which were absent in the negative control preparations, and further identification is therefore needed. In addition, the presence of two new molecules was studied in our EV fractions; CD5 was examined because it was previously present in the proteomic analyses for swine enriched EV fractions [21]. Also, the presence of CD163 (scavenger receptor and soluble extracellular form) was analyzed as it is a surface molecule related to macrophages and is thought to be involved in the ASFV viral entry pathway [43]. Both molecular markers were present in EVs with higher levels than those previously shown for swine EVs; therefore, they were selected for our further experiments. In addition, there was no difference in the elution profile in terms of protein content and FACS staining for the molecular markers, independently of the experimental group (uninfected or infected swine serum from both groups) and the virus used.

### 3.3. ASFV Proteins in EV-Enriched Fractions

The next question was whether ASFV proteins were detectable in our EV-enriched fractions using our bead-based assay and flow cytometry. Thus, the detection of CD163 and CD5 expression was combined with specific antibodies against three ASFV proteins: p30, p54 and p72.

Firstly, in order to differentiate between viral particles and EVs, viral load was evaluated in serum samples as already mentioned. Then, samples were grouped as those negative for ASFV in serum (OUR 88/3 Group A) and positive in serum (Benin ΔMGF Group B) at 24 days post-infection. As observed in Figure 3a, group A at 24 dpi (no detectable viral genomes in serum) gave measurable MFI values for viral proteins evaluated in EV-enriched fractions which decreased as later fractions were eluted from the size exclusion column. Those markers were not present in the control sera from the abattoir (Figure 3b). P72 was the viral protein with the highest MFI value in almost all EV-enriched preparations.

On the other hand, samples from group B at 24 dpi (with detectable viremia) showed positive MFI values, as observed in Figure 3c. This result was expected due to the presence of DNA, which was suggestive of the presence of viral particles in the serum samples that could be captured in the assay. As seen with OURT88/3, p72 is the viral protein with the highest MFI values of all three evaluated proteins. Importantly, control samples (negative sera) showed MFI values for exosome markers but not for viral proteins, confirming the specificity of antibodies used in this assay (Figure 3b).

### 3.4. Proteomic Analyses of ASFV-Infected Serum-Derived EVs

Protein composition in serum-derived EV fractions was analyzed in terms of ASFV viral proteins and possible changes in pig proteins due to infection. Firstly, independent groups (Group A and Group B) were compared with controls, and only viral proteins identified with 1% FDR and two or more unique peptides and which were absent in controls were marked as positive identifications. As seen in Table 1 and Table 2, some viral proteins were identified in EV-enriched fractions, although only for the Benin ΔMGF viral strain.

In Group A (infected with OURT 88/3), there were no viral proteins identified by mass spectrometry following the exclusion criteria applied to the analyses. Later, we reduced the exclusion criteria (stringency of the analysis) in the detection after 1% FDR filtering without eliminating those proteins with 1 unique peptide and evaluated those present in samples but not in controls. Following these criteria, only 3 proteins were identified (pL57L, ASFV_G_ACD_330: gene bank CBW46675.1 and VF602_ASFM2 protein B602L), two of which were in animals at 7 dpi and one in one animal at 24 dpi. Although ASFV proteins were detected by FACS (bead-based assay, Figure 3), we were not able to detect them by LC-MS/MS with the most stringent exclusion criteria.

In Group B (infected with Benin ΔMGF), EV-enriched fractions contained at least one protein identified by LC-MS/MS following the exclusion criteria, but two more proteins were detected in this preparation with less stringent analyses with 1 unique peptide and shared among samples. The best hit of this analysis was for structural protein p72 (NP_042775.1), as shown in Table 2. As viral particles were present in this group (detected by PCR), protein hits could be associated with viruses with different size ranges of EVs. P72 represented the protein with the highest MFI in our FACS analyses for EV-enriched fractions from sera of Group B (Figure 3), confirming the presence of this protein by two different methods.

When swine proteins were evaluated, a total of 942 filtered swine proteins were identified by the analysis of LFQ intensities, and some of them showed statistically significant differences in relative expression (fold change) in a virus strain-dependent manner when compared with uninfected control. Paired analyses were done between controls, groups and days post infection, showing no statistically significant differences in Group A (OURT 88/3) at 7 dpi when compared to controls and samples at 24 dpi. It is important to notice that animals of Group A (OURT-88/3) had only seven proteins in EV-enriched fractions with a significant fold change (*p* < 0.05), and from those, four were significant at *p* < 0.01: coagulation factor VIII, C-X-C motif chemokine, complement C3 (LOC100517145) and Ficolin-2 (Figure 4a and Appendix A).

Nevertheless, in Group B (Benin ΔMGF), at 7 dpi, 20 proteins were detected as differentially expressed when compared to controls (Appendix A), as well as 56 proteins in EV-enriched fractions (Figure 4b, *p* < 0.01) including HSP70, integrin beta, cAMP-dependent protein kinase, lymphocyte antigen 6 complex, fibrinogen beta chain, different types of IgG, calcium-transporting ATPase, coagulation factor VIII and a high quantity of uncharacterized pig proteins (Appendix A). In addition, when compared using a Venn diagram, all proteins identified as significant in EVs from Group A (OURT 88/3) were also identified in EV preparations from Benin ΔMGF; however, there were 49 proteins identified as unique in Group B, representing 87.5% of all significant proteins from this group (Figure 5).

Regarding EVs extracted from pigs infected with OURT88/3, there were not enough proteins to perform further analyses as only four swine proteins were differentially expressed. However, the situation for EVs from Benin ΔMGF-infected pigs was a different case, as 56 proteins from a total of 942 proteins (selected with 1% FDR and two or more unique peptides) were differentially expressed between controls and infected animals, at least in the extracellular vesicle-enriched fractions. All significant proteins were further analyzed using gene ontology and string pathways to relate expressed genes (Figure 6). The most enriched categories were threonine endopeptidase activities (molecular functions), extracellular exosome and blood microparticles (cellular components) and the integrin-mediated signaling pathway and platelet activation/aggregation (biological processes). Using the Kyoto Encyclopedia of Genes and Genomes (KEGG) analysis, the most enriched protein pathways were related to coagulation cascade and platelet activation followed by those interactions in the extracellular matrix (ECM) and focal adhesion.

## 4. Discussion

In this study, we presented the first proteomic comparison of EV-enriched fractions obtained from two different swine infections with attenuated strains of ASFV. Following previous experimental work using swine sera [21], naïve and ASFV-infected pigs’ EV-enriched fractions were evaluated with molecules reported to be present as a surface marker of swine EVs [21]. Interestingly, infection with ASFV neither altered the elution profile nor the size distribution of extracellular vesicles within enriched fractions 7 to 10 (Figure 2). Most of the vesicles measured were around 100–150 nm in diameter, but in the case of the ASFV-infected group, small, round structures were observed in some preparations (Figure 2F). We speculate that these could correspond to protein aggregates related to infection which are not seen in all infected samples (viral capsid fragments or other structures related to viral particles). Importantly, in naïve samples, those protein aggregates were not observed at all.

As ASFV infects macrophages, CD163 was included in our assays as a macrophage-related marker, complemented by CD5, which is related to T-cells and B-cells. As expected, those membrane markers allowed us to differentiate extracellular vesicle-enriched fractions independently of the virus used for infection. Although these two molecules are associated with cell membranes such as lymphocytes and macrophages, their expression after infections in extracellular vesicles derived from serum samples is variable. This is possibly associated with interactions with viral mechanisms. Interestingly, swine EVs expressed CD163 in their membranes, as detected in the bead-based assay—a result that was considered important due to the fact that CD163 was related to viral entry in ASFV [43]. However, recent work using knockout pigs lacking CD163 challenged with ASFV Georgia2007/1 were not protected from infection, demonstrating that CD163 is not crucial for viral entry, at least for this strain [44].

In our analyses of EV-enriched fractions, ASFV proteins were detected by bead-based assay in both groups using antibodies against viral proteins p30, p54 and p72. However, MFI values obtained in Group B preparations were higher than those in Group A. Those three proteins had been reported as structural proteins of early (p30/p54) and late (p72) synthesis related to viral attachment, viral internalization and capsid formation [4,45,46]. Extracellular vesicles have been reported to contain viral proteins in some studies related to ASFV in the past, but without classifying them as exosomes or extracellular vesicles [47,48,49]. In the particular case of p30, direct translocation through the plasma membrane and the release of vesicles containing virally induced proteins (structural and non-structural) has been proposed as a secretion mechanism for the extracellular transport of this protein, although unknown mechanisms have not been ruled out [49]. Suggested pathways for protein release mentioned in this work support the idea of a Golgi-independent pathway, as brefeldin treatment does not inhibit p30 release to the culture media. However, vesicles are mentioned as an alternative route for protein release, but it has been proved that exosome-like EV secretion is partially blocked by brefeldin (quantified indirectly by the expression of CD81 in Western blot, CD63-GFP expression by FACS and microRNA) [50].

Several viral proteins had been identified as potential vaccine targets and are capable of inducing neutralizing antibodies including p30, p54 and p72 [16,51]. However, antibodies for those proteins are not sufficient to protect pigs against viral replication in a challenge trial, as vaccination approaches demonstrated only a delay in the appearance of clinical symptoms and reduced levels of viremia when compared to the control group [52] but not sterilizing immunity.

Extracellular vesicles have been studied during viral infections, but nevertheless their role is still controversial depending on each particular interaction between the virus and the host. In some cases, viral proteins associated with these vesicles facilitate viral infectivity and impair the immune response in the host. Also, vesicles are able to transfer the cell receptors necessary to infect cells that were previously incapable of interacting with the virus [28,31,53,54,55]. On the other hand, it has been reported that EVs from virus-infected cells, such as HSV-1, could induce the opposite effect and inhibit viral infection by activating different immune mechanisms mediated by the delivery of viral antigens on EVs to receptor cells [33,56]. Whether ASFV-derived vesicles have a role in pathogenesis or immune response is yet to be examined.

Few viral proteins were identified by mass spectrometry in both EV preparations (Groups A and B) and only p72 was selected by our exclusion criteria of two unique peptides. Nevertheless, other viral proteins were detected in more than one animal, but with 1 unique peptide, making those identifications potential candidates for further evaluation. We suspect that the small number of proteins identified could be related to a lack of information regarding protein sequences for ASFV virus. It is important to mention that the ASFV protein database available at UniprotKB has presently 4419 protein sequences—a small number compared with other viruses such as PRRSV (around 25,000 sequences) or influenza virus (788,906 sequences). Another limitation in protein identification is associated directly with biosafety protocols for analyzing BSL-3/SAPO4 samples of ASFV. In our case, EV-enriched fractions required heat inactivation at 60 °C for 2 h. This process could contribute to protein degradation and loss of information within EV-enriched samples, because when evaluated after EV isolation for viral proteins in the FACS bead-based assay, at least some samples gave positive results in the assay using specific antibodies against those proteins. The different thresholds in each technique could be related to this apparent discrepancy in protein detection when comparing both sets of data. Taken together, these results indicate that EVs could be a source of new viral antigens that could explain pathogenicity and immune response during infection with different viral strains, but methods for sample processing in BSL-3/SAPO4 containment need to be improved to increase protein hits in the analysis.

In addition, it was necessary to evaluate swine proteins on EV cargo and determine whether viral infection with different strains could modify the protein content of those EVs. Interestingly, we identified 942 swine proteins in all EV preparations (controls, and OURT 88/3 and Benin ΔMGF-infected preparations); however, in OURT 88/3 swine sera, only a small number of proteins were differentially expressed compared to controls. OURT 88/3 has been reported to be a low virulent strain (naturally attenuated) with few to no clinical signs and no detectable/low intermittent viremia post immunizations [20], and this low pathogenicity with almost no effect in the host could reflect the few modifications of porcine protein expression detected in the EV-enriched fractions when compared with control (non-infected) samples. Importantly, infections with this attenuated strain protect against a lethal challenge from the same genotype I and even other genotypes [17,57]. In contrast, Benin ΔMGF is a deletion mutant in which some genes of the multigene family (related to IFN expression) were deleted. These genes were similar to but not the same as those absent in OUTR 88/3. Benin ΔMGF has intact CD2v and EP153R genes as compared to OURT88/3. Animals infected with this mutant exhibit detectable viremia and clinical signs post-infection, possibly associated to another feature of this particular strain; despite the absent genes in both viruses [20], nevertheless, this mutant has been characterized as an inducer of protection [42].

For all these reasons, we tentatively hypothesize that there was a relationship between the viral proteins detected in the EV-enriched fractions we examined and the modifications in swine proteins associated with those fractions of EVs (20 significant proteins at 7 dpi and 56 significant proteins at 24 dpi), including those proteins identified as differentially expressed (seven proteins) in OURT 88/3 infections. Most of these proteins are related to threonine endopeptidase activities, integrin and fibrinogen complexes and platelet activation and integrin-mediated signaling pathways. Previous reports about proteomic analysis after infection have been performed using either Vero cells or in vivo trials using gastro-hepatic lymph nodes exhibiting similar protein groups which were identified as over or sub-expressed, such as proteasome-related proteins, heat shock proteins or immunoglobulins [58,59], despite our focus on only extracellular vesicle-enriched fractions. Importantly, those results open a new path to explore the interactions between host cells and the virus, including the modulation of the immune system by different strategies including extracellular vesicles. For example, platelet function abnormalities or thrombocytopenia have been related to unpredictable delivery hemorrhages in inherited disorders in humans [60]. Also, fibrinogen is a serum multi-chain protein which, when activated, aggregates to form fibrin, one of the main components of a blood clot [61]. On the other hand, it has been shown that dengue virus nonstructural-1 protein (NS1) generates antibodies to common epitopes on human blood clotting and integrin/adhesin proteins, with potential implications in hemorrhagic fever pathogenesis being induced by this virus [62]. All in all, these findings suggested some alterations in the coagulation system that could be associated with the virus–host cell interactions involved in the pathogenesis of ASFV. Further investigations are required to sustain this hypothesis.

## 5. Concluding Remarks

In this study, we presented new evidence that different strains of ASFV modify one aspect of host biology related to extracellular vesicles. Several viruses are able to exploit the EV formation and secretion pathway in different ways to either interfere with immune responses [28,31] or block viral entry and replication [33,56]. We think that exploring new aspects of ASFV infections such as EV-enriched fraction content (viral and host proteins) will contribute to understanding how different viral strains interact with host cells and modify the expression of immune response genes and proteins. Recent research works have made contributions here by exploring the proteomics of the viral particles and intracellular proteome of this virus [3,63]. Our work was focused on extracellular vesicle-enriched fractions, as these have been shown to be important for several viral infections, and recently they have been used as a vaccination strategy for another important swine disease—PRRS virus [22].

As mentioned, ASFV is a complex virus that is causing major outbreaks in Europe and Asia [12,13,64]. With no vaccine or treatment available for this pathogen, slaughter is the only effective control measure [65]. Several genotypes [66] and immune-suppression genes [67], long-term viral infections [68,69], gaps in knowledge in biology and immunology and several spreading vectors such as wild boars and *Ornithodoros spp.* ticks [70,71,72,73] generate an urgent need to cover the less-explored aspects of this virus to control or prevent pandemics that could affect the global economy in animal production.

## Figures and Tables

**Figure 1 viruses-11-00882-f001:**
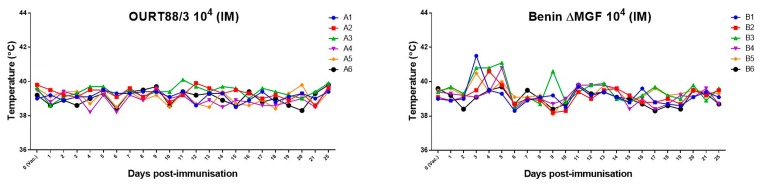
Rectal temperatures in pigs. Rectal temperatures were assessed at different days after immunization with two different African swine fever virus (ASFV) strains, (**a**) OURT-88/3 and (**b**) deletion mutant BeninDMFG, by an intramuscular route. Before inoculation, rectal temperatures of pigs ranged between 39 and 39.9 °C. At day 24 post immunization, the rectal temperatures of pigs immunized with OURT88/3 ranged between 39.4 and 39.9 °C, while in the group of pigs immunized with Benin ΔMGF, temperatures ranged between 38.7 and 39.6 °C.

**Figure 2 viruses-11-00882-f002:**
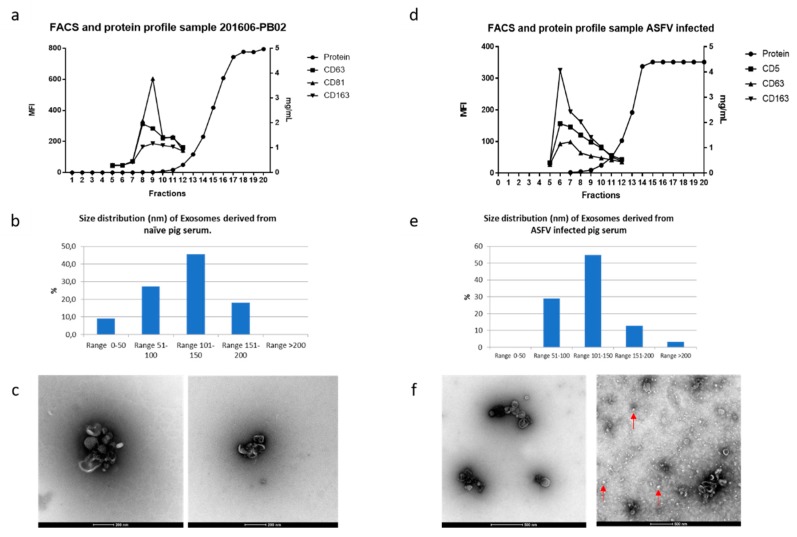
Characterization of extracellular vesicles derived from uninfected and ASFV infected swine sera. (**a**,**d**) FACS analysis of extracellular vesicle (EV) fractions derived from infected and control animals; (**b**,**e**) size distribution as measured by electron microscopy; (**c**,**f**) transmission electron microscopy of negatively-stained uninfected and infected samples, respectively. EVs are shown with red arrows (scale bar in c: 200 nm; scale bar in f: 500 nm).

**Figure 3 viruses-11-00882-f003:**
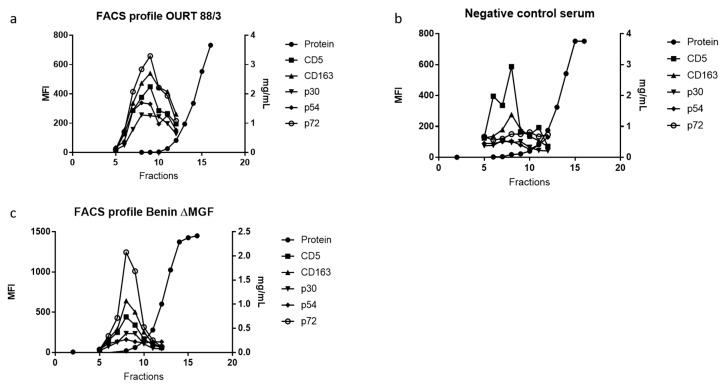
FACS analysis of EV-enriched fractions derived from infected and control animals. (**a**) EVs from representative sera of an infected animal with OURT 88/3 at 24 days post infection, (**b**) EV profiles from a representative of uninfected swine sera, (**c**) EV FACS profile from a representative animal infected with Benin ΔMGF virus. In all samples, molecular markers CD5 and CD163 were used as control markers for EVs and three different viral proteins were evaluated (p30, p54 and p72).

**Figure 4 viruses-11-00882-f004:**
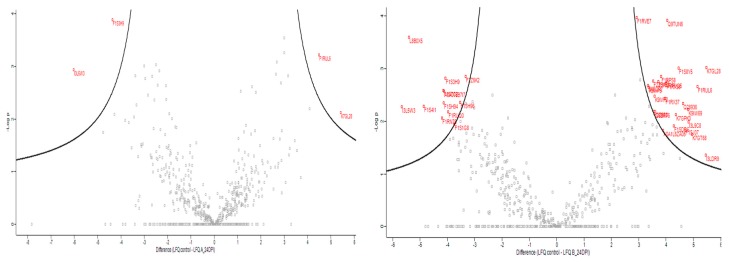
Swine proteins on EV-enriched fractions evaluated at 24 days post infection (red proteins were significant at *p* < 0.05) and above the volcano plot threshold (*p* < 0.01). (**a**) Proteins differentially expressed in EV-enriched fractions from Group A (infected with OURT 88/3). (**b**) Proteins differentially expressed in EV-enriched fractions from Group B (infected with Benin ΔMGF).

**Figure 5 viruses-11-00882-f005:**
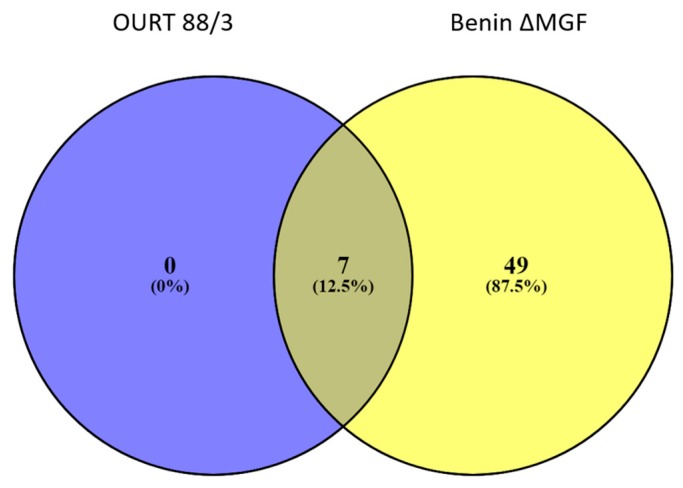
Venn diagram for the comparison of significantly expressed swine proteins in a virus-dependent manner detected in serum-derived EVs.

**Figure 6 viruses-11-00882-f006:**
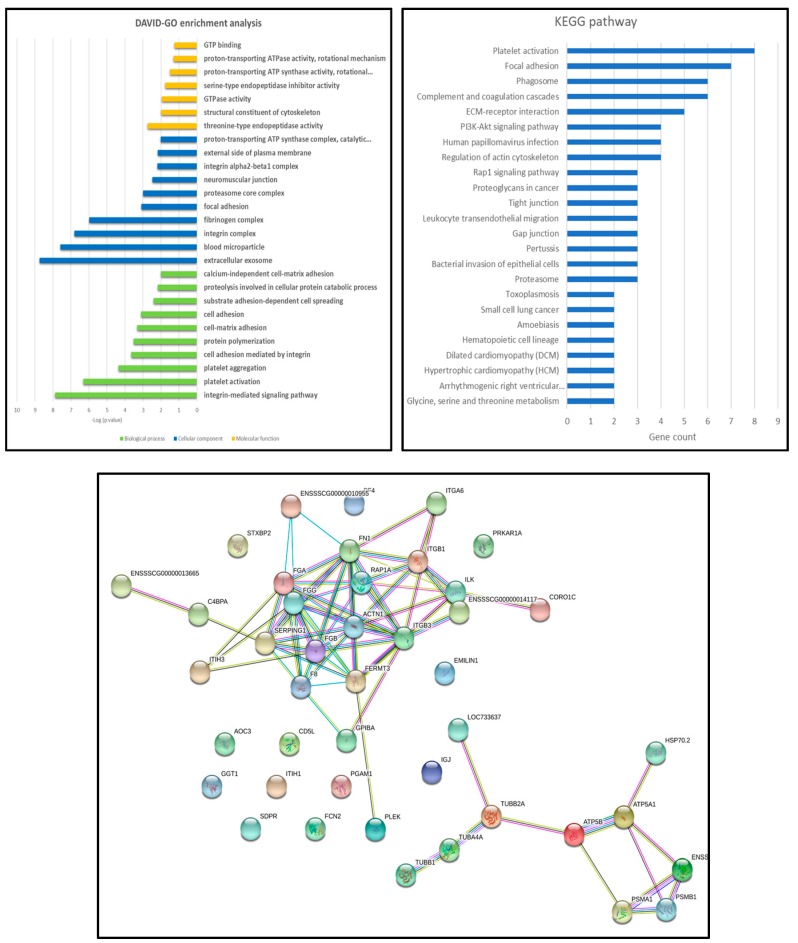
Gene ontology analyses of Benin ΔMGF EV-enriched fractions, where differentially expressed swine proteins were classified according to the biological process, cellular component or molecular function, as well as a string pathway analysis for all proteins differentially expressed in Benin ΔMGF-infected animals.

**Table 1 viruses-11-00882-t001:** Identified viral proteins in the OURT 88/3 group of porcine serum-derived EV-enriched fractions at 24 days post infection. Columns correspond to each animal at different time points (Group A, 7 and 24 dpi) where unique peptides were identified (red).

Identified Protein (by Unique Peptides) and Pig ID with Days Post Infection	A1D7	A2D7	A1D24	A2D24	A4D24	A5D24	A6D24	CONTROL 1	CONTROL 2
pL57L	0	1	0	0	0	0	0	0	0
ASFV_G_ACD_00330 (gene bank CBW46675.1)	0	1	0	0	0	0	0	0	0
VF602_ASFM2 Protein B602L	0	0	1	0	0	0	0	0	0

**Table 2 viruses-11-00882-t002:** Identified viral proteins in the Benin ΔMGF group of porcine serum-derived EV-enriched fractions at 24 days post infection. Columns correspond to each animal at different time points (Group B, 7 and 24 dpi) where unique peptides were identified (red).

Identified Protein (by Unique Peptides) and Pig ID with Days Post Infection	B1D7	B2D7	B1D24	B2D24	B3D24	B4D24	B5D24	B6D24	CONTROL 1	CONTROL 2
gi|858945434|gb|AKO62698.1| pJ328L	0	0	0	0	0	1	0	1	0	0
Structural protein p72	1	1	1	2	0	0	0	1	0	0
Hhypothetical protein AFSV47Ss_0158	1	1	0	1	0	1	0	1	0	0
VF354_ASFWA Uncharacterized protein B354L	0	1	1	0	1	0	0	0	0	0

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
