# Peer review of "Serum-Derived Extracellular Vesicles from African Swine Fever Virus-Infected Pigs Selectively Recruit Viral and Porcine Proteins"

_viruses, 2019, doi:10.3390/v11100882_

Round 1
Reviewer 1 Report
Montaner-Tarbes et al characterize serum-derived extracellular vesicles (EVs) from African Swine Fever virus (ASFV)-infected pigs. EVs collected from pigs infected with strains OURT 88/3 and Benin ΔMGF were characterized by surface marker expression, electron microscopy and proteomic analysis. The authors found that EVs contain ASFV proteins and variable number of host proteins. Overall, the article is well written and figures are appropriate. A few comments below.
Fig. 3. Are values in the figure means for the groups? Clarify in legend. if so, deviation bars should be included.
Also the legend should read “FACS analysis of EV fractions derived from infected and control animals” instead of “infected EV fractions”.
Fig 2f legend. Use arrows to indicate “unidentified” structures associated with viral infection.
L155, spell SEC.
L155-160. Looks like this paragraph is repeated twice.
L245 and 249. 2a, 2c, 2d, 2f.
L288 and 314. Replace “viral particles” with viral DNA suggestive of presence of viral particles.
Table I. what viral protein is indicated by ASFV_G_ACD_00330?
Fig. 6; results for GO analysis are barely readable.
- L402-406 look unrelated to the manuscript topic.
- As stated by the authors, heat inactivation of EV-enriched fractions might have impacted the results. Comparison of normal serum profiles with and without heat inactivation should provide a measure of this effect.
- The different outcomes with flow cytometry and LC-MS deserve a comment.
- Tables I and II, The box on the top left should indicate both proteins and pig IDs.
Author Response
Reviewer 1.-
Fig. 3. Are values in the figure means for the groups? Clarify in legend. if so, deviation bars should be included.
A: The values corresponded to a single sample, as a representative of the whole group. Following the reviewer´s comment, the figure legend has been modified accordingly.
Also the legend should read “FACS analysis of EV fractions derived from infected and control animals” instead of “infected EV fractions”.
A: Following the reviewer´s comment, the figure legend has been modified accordingly.
Fig 2f legend. Use arrows to indicate “unidentified” structures associated with viral infection.
A: Following the reviewer´s comment, figure 2 has been modified accordingly.
L155, spell SEC.
A: Following the reviewer´s comment, SEC has been modified accordingly.
L155-160. Looks like this paragraph is repeated twice.
A: Paragraph removed. Apologies
L245 and 249. 2a, 2c, 2d, 2f.
A: All figure letters have been modified accordingly.
L288 and 314. Replace “viral particles” with viral DNA suggestive of presence of viral particles.
A: The text has been modified accordingly.
Table I. what viral protein is indicated by ASFV_G_ACD_00330?
A: This information has already been included in the text with its Gene bank accession number.
Fig. 6; results for GO analysis are barely readable.
A: The figure has been modified to improve it.
- L402-406 look unrelated to the manuscript topic.
A: Paragraph removed. Apologies
- As stated by the authors, heat inactivation of EV-enriched fractions might have impacted the results. Comparison of normal serum profiles with and without heat inactivation should provide a measure of this effect.
A: We thank the reviewer for this suggestion which will have already thinking. We will perform in the future experiments comparing different inactivation protocols to be used inside high containment facilities to facilitate sample processing and to obtain the best possible information.
- The different outcomes with flow cytometry and LC-MS deserve a comment.
A: We thank the reviewer for this suggestion and some sentences were added about this point in the 6th paragraph of the discussion.
- Tables I and II, The box on the top left should indicate both proteins and pig IDs.
A: The tables have been modified accordingly.

Reviewer 2 Report
The authors present a very novel experiment to identify viral proteins and host proteins differentially express after ASFV infection. As far as I know this is the first time an experiment like this is carried out. ASFV is a devastating disease and a lot more basic science like this is needed to understand the pathology of the infection/disease and how to address solutions. The technology used in the experiment is very novel as well in the ASFV field and could open new avenues for future experiment analysis.
Minor comments
All manuscript. The authors refer to BeninDMFG and BeninDMGF. There is a need for harmonizing the nomenclature.
Abstract
Line 29: the authors should add “virus” in the following sentence “The causative agent, African swine fever (ASFV)”
Introduction
Line 41: The authors should remove or the word “swine” or the word ‘suids”. It becomes redundant.
Line 101: “Given the role of extracellular vesicles in inducing immunological responses in pigs”. It is not clear from the introduction what is this role in pigs. The authors give some examples about the importance of EVs in some diseases but there are not pig related references.
Material and methods
Line 118-119: “Samples were taken from al larger experiment (unpublished results) from day 0 to day 24.” It is not relevant for the manuscript to know the samples come from a larger experiment. I would suggest removing this sentence or re-write it.
Line 145 and 159 and 180: the authors should not underline the temperature. E.g. “at 4ºC”. it should be “at 4°C”.
Line 147 and 165: “dilution of secondary antibody FITC”. The authors should specify the specificity of the antibodies. E.g. anti-mouse FITC.
Line 149: “were suspended 4% paraformaldehyde”. I think it should be “were resuspended with 4% paraformaldehyde”
Line 150: “Then, plate was”. I think it should be “Then, the plate was…”
Line 153: the FlowJo software version is missing.
Line 154-155: harmonize “bead based assay” or “bead-based assay”.
Line 160-165: duplication of the previous text “A bead-based assay for detection of viral proteins was used to phenotypically identify SEC fractions containing EVs and viral proteins [19]. Briefly, EVs enriched fractions were coupled with Aldehyde/Sulfate Latex Beads, 4% w/v, 4 μm (Invitrogen) and then blocked with PBS 1X / BSA 0.1% (Sigma-Aldrich) /NaN3 0.01% (Sigma-Aldrich). Fractions were incubated in microtest round bottom 96-well plates for 30 minutes at 4ºC with anti-p30, anti-p54 and anti-p72 antibodies (The Pirbright Institute [15,35,36] at 1:100 dilution.”
Line 178: the software version is missing.
Line 185: please, specify what TCA stands for.
Line 189: “ranged from 9.8 × 107 to 3.9 × 108 particles/mL”. I think the authors are missing the superscript. “9.8 × 107to 3.9 × 108particles/mL”.
Line 223: the authors should consider adding a reference for the STRING software, as indicated in the STRING software webpage.
Results
Line 228: the authors should not underline the temperature. E.g. “at 40.5ºC”. it should be “at 40.5°C”.
Figure 1: the authors should consider increasing the font size in the X and Y axes in both panels.
Line 235: “at day 7dpi showed”. It should be “at day 7 pi showed”.
Line 240-242: please re-phrase this paragraph, it is difficult to follow. “As some EVs and viral particles have similar size, it was relevant to establish whether ASFV could be present in extracellular vesicles enriched fractions to avoid possible false identifications of viral proteins in further analyses.”
Figure 2 panel a and d: please, harmonize the markers used in the assay. Panel A: CD63, CD81 and CD163 and in panel D: CD5, CD63 and CD163. And also please, harmonize the tittle of all panels, font size and legends.
Discussion
Line 406: the reference should go at the end of the sentence.
Major comments
The proteins identified by the proteomic analysis (section 3.4) are very little. Maybe the assays need some improvement because only p72 was identified. And the criteria were not very stringent. I really think the authors should consider merging section 3.3 with 3.4.
I would strongly suggest the authors to include the differential protein expression in an active infection compared to a cleared infection. Maybe the authors should consider analyze 7 dpi (after peak of viremia, maybe?) vs 24 dpi (less virus in blood or cleared, I guess) with the same virus strain. Now, it is not possible to know if the different expression is due to different virus strains or due to the level of viremia. It would be also nice to include the virus titers in the manuscript. If the authors cannot perform this analysis, I would suggest for them to go for a short communication rather than full paper.
At least two other papers were talking about protein expression during infection. Nothing was discussed in this sense. Comparative proteomic analysis reveals different responses in porcine lymph nodes to virulent and attenuated homologous African swine fever virus strains.Herrera-Uribe J, Jiménez-Marín Á, Lacasta A, Monteagudo PL, Pina-Pedrero S, Rodríguez F,MorenoÁ, Garrido JJ. Vet Res. 2018 Sep 12;49(1):90. doi: 10.1186/s13567-018-0585-z. And Identification of cellular proteins modified in response to African swine fever virus infection by proteomics.Alfonso P, Rivera J, Hernáez B, Alonso C, Escribano JM. Proteomics. 2004 Jul;4(7):2037-46.
Author Response
Reviewer 2.-
All manuscript. The authors refer to BeninDMFG and BeninDMGF. There is a need for harmonizing the nomenclature.
A: The manuscript has been reviewed throughout and all the references to this virus changed to Benin DMGF.
Abstract
Line 29: the authors should add “virus” in the following sentence “The causative agent, African swine fever (ASFV)”
A: Done
Introduction
Line 41: The authors should remove or the word “swine” or the word ‘suids”. It becomes redundant.
A: Done
Line 101: “Given the role of extracellular vesicles in inducing immunological responses in pigs”. It is not clear from the introduction what is this role in pigs. The authors give some examples about the importance of EVs in some diseases but there are not pig related references.
A: the sentence has been modified accordingly.
Material and methods
Line 118-119: “Samples were taken from al larger experiment (unpublished results) from day 0 to day 24.” It is not relevant for the manuscript to know the samples come from a larger experiment. I would suggest removing this sentence or re-write it.
A: the sentence has been modified accordingly.
Line 145 and 159 and 180: the authors should not underline the temperature. E.g. “at 4ºC”. it should be “at 4°C”.
A: the symbol has been modified accordingly.
Line 147 and 165: “dilution of secondary antibody FITC”. The authors should specify the specificity of the antibodies. E.g. anti-mouse FITC.
A: the sentence has been modified accordingly.
Line 149: “were suspended 4% paraformaldehyde”. I think it should be “were resuspended with 4% paraformaldehyde”
A: the sentence has been modified accordingly.
Line 150: “Then, plate was”. I think it should be “Then, the plate was…”
A: the sentence has been modified accordingly.
Line 153: the FlowJo software version is missing
A: the sentence has been modified accordingly stating that FlowJo vX for PC TreeStar was used.
Line 154-155: harmonize “bead based assay” or “bead-based assay”.
A: The expression “bead based assay” has been harmonized in the manuscript.
Line 160-165: duplication of the previous text “A bead-based assay for detection of viral proteins was used to phenotypically identify SEC fractions containing EVs and viral proteins [19]. Briefly, EVs enriched fractions were coupled with Aldehyde/Sulfate Latex Beads, 4% w/v, 4 μm (Invitrogen) and then blocked with PBS 1X / BSA 0.1% (Sigma-Aldrich) /NaN3 0.01% (Sigma-Aldrich). Fractions were incubated in microtest round bottom 96-well plates for 30 minutes at 4ºC with anti-p30, anti-p54 and anti-p72 antibodies (The Pirbright Institute [15,35,36] at 1:100 dilution.”
A: the sentence has been modified accordingly.
Line 178: the software version is missing.
A: the sentence has been modified accordingly stating that FlowJo vX for PC TreeStar was used.
Line 185: please, specify what TCA stands for.
A: the sentence has been modified accordingly.
Line 189: “ranged from 9.8 × 107 to 3.9 × 108 particles/mL”. I think the authors are missing the superscript. “9.8 × 107to 3.9 × 108particles/mL”.
A: the sentence has been modified accordingly.
Line 223: the authors should consider adding a reference for the STRING software, as indicated in the STRING software webpage.
A: the sentence has been modified accordingly.
Results
Line 228: the authors should not underline the temperature. E.g. “at 40.5ºC”. it should be “at 40.5°C”.
A: the sentence has been modified accordingly.
Figure 1: the authors should consider increasing the font size in the X and Y axes in both panels.
A: The quality of the image has improved and figure 1 has been resubmitted in the revised version.
Line 235: “at day 7dpi showed”. It should be “at day 7 pi showed”.
A: the sentence has been modified accordingly.
Line 240-242: please re-phrase this paragraph, it is difficult to follow. “As some EVs and viral particles have similar size, it was relevant to establish whether ASFV could be present in extracellular vesicles enriched fractions to avoid possible false identifications of viral proteins in further analyses.”
A: the paragraph has been modified accordingly for the sake of clarity.
Figure 2 panel a and d: please, harmonize the markers used in the assay. Panel A: CD63, CD81 and CD163 and in panel D: CD5, CD63 and CD163. And also please, harmonize the tittle of all panels, font size and legends.
A: panels have been harmonised accordingly.
Discussion
Line 406: the reference should go at the end of the sentence.
A: the reference has been modified accordingly.
Major comments
The proteins identified by the proteomic analysis (section 3.4) are very little. Maybe the assays need some improvement because only p72 was identified. And the criteria were not very stringent. I really think the authors should consider merging section 3.3 with 3.4.
A: as discussed in the manuscript, we think that that small number protein identification could be related to lack of information of protein sequences for ASFV virus. It is important to mention that ASFV protein database available at UniprotKB has presently 4419 protein sequences, a small number compared with other viruses such as PRRSV (around 25000 sequences) or Influenza virus (788906 sequences). Another limitation in protein identification is associated directly to biosafety protocols for analyzing BSL-3/SAPO4 samples of ASFV. In our case, EV enriched fractions required heat inactivation at 60°C for 2 hours. This process could contribute to protein degradation and loss of information within EV enriched samples, because when evaluated after EVs isolation for viral proteins in the FACS bead-based assay, at least some of them gave positive results in the assay using specific antibodies against those proteins. Different threshold in each technique could be related to this apparent discrepancy in protein detection when comparing both sets of data.
I would strongly suggest the authors to include the differential protein expression in an active infection compared to a cleared infection. Maybe the authors should consider analyze 7 dpi (after peak of viremia, maybe?) vs 24 dpi (less virus in blood or cleared, I guess) with the same virus strain. Now, it is not possible to know if the different expression is due to different virus strains or due to the level of viremia. It would be also nice to include the virus titers in the manuscript. If the authors cannot perform this analysis, I would suggest for them to go for a short communication rather than full paper.
A: we thank the reviewer for this suggestion which we widely discussed during the experimental planning. However, due to possible virus replication at day 7 pi, those samples could not be used to evaluate viral protein content of EVs as virus will mask those proteins associated to them. However, samples were checked at 14 and 24dpi and none of the pigs immunized with OURT88/3 (group A) showed detectable levels of virus genome in blood, while in all pigs immunized with Benin ΔMGF (group B) moderate levels of ASFV DNA were detected as ranging from 2.85x104 to 1.45x105 copies at 14dpi and 4.03x103 to 2x104 copies at 24dpi. Also, we decided to use those attenuated ASFV strains because when using a virulent strain, pigs have to be culled for humane reasons around day 5 post challenge (see Sanchez-Cordon et al, Antiviral Research 2017, doi: 10.1016/j.antiviral.2016.11.021). Taking samples within this window of days would imply using samples positive for ASFV DNA, which again could mask our results.
At least two other papers were talking about protein expression during infection. Nothing was discussed in this sense. Comparative proteomic analysis reveals different responses in porcine lymph nodes to virulent and attenuated homologous African swine fever virus strains.Herrera-Uribe J, Jiménez-Marín Á, Lacasta A, Monteagudo PL, Pina-Pedrero S, Rodríguez F,MorenoÁ, Garrido JJ. Vet Res. 2018 Sep 12;49(1):90. doi: 10.1186/s13567-018-0585-z. And Identification of cellular proteins modified in response to African swine fever virus infection by proteomics. Alfonso P, Rivera J, Hernáez B, Alonso C, Escribano JM. Proteomics. 2004 Jul;4(7):2037-46.
A: We thank the reviewer for pointing to these papers that we have included in the discussion section in the manuscript.

Reviewer 3 Report
In this manuscript, the authors describe the serum-derived extracellular vesicles from ASFV infected pig selectively recruit viral and porcine proteins. The authors give an overall good background on ASFV and extracellular vesicles in viral infection. This is a valuable research about the pathogenesis in ASFV infection.
Major comments:
Line 56-57. …by the first cases of ASFV on June 2018, in to the world’s largest pig producer, China [11, Zhou et al., 2018 Transboundary and Emerging Diseases https://doi.org/10.1111/tbed.12989]. Line 28 Vietnam [Le et al., 2019 Emerging Infectious Diseases 26:1433-1435]. ℃is different throughout the manuscript, for example, line 145, 148, 159, 166, 228, 425. Line 119, al larger experiment. What is al ? Figure 1 A and B, give a reference interval of body temperature of 9-10-week-old pigs. Figure 3 legend, days post infection rather than dpi. line 45 “180-200nm” change to “180-200 nm”; line 48 “193kbp” change to 193 kbp. Authors need to check throughout the text.
Author Response
Reviewer 3
Line 56-57. …by the first cases of ASFV on June 2018, in to the world’s largest pig producer, China [11, Zhou et al., 2018 Transboundary and Emerging Diseases https://doi.org/10.1111/tbed.12989].
Line 28 Vietnam [Le et al., 2019 Emerging Infectious Diseases 26:1433-1435].
A: those references have been included in the introduction
℃is different throughout the manuscript, for example, line 145, 148, 159, 166, 228, 425. Line 119, al larger experiment.
A: this symbol has been modified in the manuscript throughout.
What is al ?
A: Corrected in the manuscript.
Figure 1 A and B, give a reference interval of body temperature of 9-10-week-old pigs.
A: as suggested by the reviewer, a reference with the interval of body temperatures has been added to Figure 1 legend.
Figure 3 legend, days post infection rather than dpi.
A: Following the reviewer´s comment, figure 3 legend was modified.
line 45 “180-200nm” change to “180-200 nm”; line 48 “193kbp” change to 193 kbp. Authors need to check throughout the text.
A: Following the reviewer´s comment, “nm” and “kbp” was checked throughout the manuscript.
